# Every day mitral valve reconstruction: What has changed over the last 15 years?

**Farnoosh Motazedian, Roya Ostovar[ID], Martin Hartrumpf, Filip Schröter, Johannes M. Albes[ID] \***

Department of Cardiovascular Surgery, Heart Center Brandenburg, University of Brandenburg Medical School Theodor Fontane, Neuruppin, Germany

\* Johannes.albes@immanuelalbertinen.de

## Abstract

### Objective

Mitral valve reconstruction (MVR) is one of the cardiosurgical procedures which cannot be substituted by any intervention owing to the quality of the quasi-anatomical, physiological repair. However, technique and strategies have changed over the years. We looked at procedural characteristics and outcome in an all-comer, non-selected cohort of patients.

### Methods

738 out of 1.977 patients were retrospectively analyzed receiving MVR with and without concomitant procedures. The cohort was divided into three periods. P1: 2004–2009 (134 pts.); P2: 2010–2014 (294 pts.), and P3: 2015–2019 (310 pts.).

### Results

Early mortality increased from P1 to P2 and decreased from P2 to P3 (9% P1, 13% P2, 10% P3). All patients received an annuloplasty-ring. In P1 resection measures dominated. In P3 artificial chordae were dominant. Age, BMI, and risk scores correlated with early mortality. Survival rates were 66% (5-years), 55% (10-years), 44% (15-years) in P1, 63% (5-years), 50% (10-years) in P2, and 80% (5-years) in P3. Odds ratio for reduced long-term survival were concomitant venous only bypass surgery (10-years 2,701, p = 0.026). 10-year survival was positively influenced by isolated MVR (0.246, p = 0.001), concomitant isolated arterial bypass (IMA) (0.153, p = 0.051), posterior leaflet measure (0.178, p<0.001), and use of artificial chordae (5-years 0.235, p<0.001).

### Conclusion

Indication for ring implantation remained mandatory while preference changed alongside improved designs. Procedural characteristics changed from mainly resection maneuvers to predominant use of artificial chordae. Long-term results were negatively influenced by comorbidities and positively influenced by posterior leaflet repair and artificial chordae. MVR underwent a qualitative evolution and remains a valuable cardiosurgical procedure.

**Data Availability Statement:** The data protection officer of the Brandenburg Medical School Theodor Fontane determined that patient data can only be shared publicly in a strictly anonymized fashion because of publication restrictions by current data

protection laws in Europe. The anonymized data set is available in DRYAD at doi:10.5061/dryad.4f4qrfjf1.

**Funding:** The authors received no specific funding for this work.

**Competing interests:** The authors have declared that no specific interests exist.

## Introduction

Mitral valve reconstruction (MVR) is one of the remaining cardiac surgical procedures that, due to the quality of the quasi-anatomical and thus physiological repair, cannot be replaced by any intervention. In the 1970s, Alain Carpentier introduced the systematic approach to mitral valve reconstruction into the clinical practice. The clarification of the underlying pathology types was accompanied by appropriate measures to treat them and thus to reconstruct the valve as anatomically and physiologically as possible [1–3]. This was clearly a breakthrough that far surpassed existing mitral repair maneuvers and thus opened the future of surgical repair not only of the mitral valve, but later also of the tricuspid and aortic valves. Today between half and two thirds of the mitral valves are being reconstructed in countries with an efficient health system [4]. Although the techniques invented and refined by Carpentier are still widely used, mitral valve reconstruction has gone through an evolutionary process over the years in which some techniques became obsolete or even died out while others flourished. We therefore examined both procedural features and outcome in an unselected group of patients who received MVR in our institution over the past 15 years.

## Patients and methods

Of a total number of 1.977 mitral valve procedures 738 patients were retrospectively analyzed receiving MVR with and without any concomitant procedure from 2004 to 2019. Underlying pathologies leading to mitral regurgitation necessitating repair were distinguished between primary degenerative and secondary functional (ischemic) origin according to the current ESC-guidelines [5]. Patients who had to undergo a cardiac tumor resection were excluded, even if this tumor could be removed from the mitral valve and the valve left in place. Also excluded were patients with acute endocarditis, which led to resection measures on an otherwise preserved mitral valve. Furthermore, patients were excluded who did not want to participate in the study or they or their relatives were unable to undergo informed consent. Ethics vote was obtained by the Ethics Committee of our university. Owing to the entirely retrospective nature of the study, the necessity for written informed consent was waived. Late survival state was obtained in several cases by telephone contact. That was considered as oral and thus adequate consent by the ethics committee. (Sept. 2020; File No. E-01-20200709). The cohort was divided into three periods: Period one (P1): 2004–2009 (134 pts.); period two (P2) 2010–2014 (294 pts.); and period three (P3) 2015–2019 (310 pts.). Long-term results up to 15 years were obtained. Statistical evaluation was performed using Microsoft Excel 2010 and R® [6]. The descriptive statistics initially included all demographic items as well as perioperative parameters recorded during the patient stay and presented here as a mean value with standard deviation (95% confidence interval). Numerical data was tested for normal distribution before being compared with Student's t-test or Mann-Whitney-U Test respectively. Categorical data was compared using Fisher's exact test and Chi-squared test. To examine the existence of trends in categorical variables over the three time periods, Cochran-Armitage test for trend in proportions was used. In addition, a risk factor analysis was carried out to determine the odds ratio. Kendall's Tau was used to correlate hospitalization time and a panel of typical risk factors. Differences were assumed to be significant if $p < 0.05$. The patients were contacted by phone for follow up. No information could be obtained from 55/738 patients (93% follow-up). The survival times were calculated using the Kaplan-Meier method. The primary endpoint was all-cause mortality, the secondary endpoints were: recurrence rate requiring reoperation, early mortality and long-term survival in the years 2004–2009 (Period 1); 20010–2014 (Period 2); 2015–2019 (Period 3). Additional secondary endpoints were the type of surgery and the

type and scope of other simultaneous surgical measures and their influence on early mortality and long-term survival.

## Results

The proportion of mitral valve repair from all mitral valve procedures more than doubled from period 1 (22.1%) to period 2 (47.6%) and slightly decreased thereafter in period 3 (41.1%). Mitral valve pathology was of primary nature in 64,6% of all patients. Age remained quite stable (P1: 64.26 years, P2: 66.46 years, P3: 65.42 years), risk scores (EuroScore (ES), logistic ES (log. ES)) increased from P1 to P2 and decreased thereafter. Mean hospitalization time decreased (17.7 days P1; 17.5 days P2; 15.2 days P3) (Table 1).

Number of isolated procedures varied between 30 and 50% (48% P1, 31% P2, 40% P3). Minimal-invasive MVR did not play a major role in this non-selected cohort with only 13 patients mainly carried out in period 2. Early mortality increased from P1 to P2 and decreased from P2 to P3 (9% P1, 13% P2, 10% P3). Concomitant procedures were mostly coronary artery bypass graft as well as aortic valve replacement and endocardial cooled radiofrequency thermal atrial ablation (MAZE) was often performed (Table 2).

All patients in all periods received an annuloplasty-ring. While Edwards Classic™ (35.07%) and Livanova (previously Sorin™) Anuloflo™ (51.49%) were mostly used in P1 it gradually shifted until P3 to the Edwards Physio II (65.16%) and the St. Jude Rigid Saddle (25.48%). In P1 resection measures for the posterior leaflet as well as Wooler plasty dominated. In P2 artificial chordae came increasingly in use while resection measures decreased. In Period 3 artificial chordae were the dominant procedure (Table 3, Fig 1).

High age, BMI, and risk scores correlated with early mortality (EM) (Age-EM: 71.54 Age-Control: 64.89%, $P<0.001$; BMI-EM: 28.12% BMI-Control: 26.72, $P = 0.026$; log. ES-EM: 30.98%, Log.ES-Control: 10.46%, $P<0.001$; ESII-EM: 16.99% ESII-Control: 4.63%, $P<0.001$). Concomitant bypass surgery, particularly without IMA as well as concomitant implantation or explantation of electrophysiological devices increased the risk for early mortality whereas isolated MVR and use of artificial chordae reduced it (Table 4).

Freedom from early or late redo-procedures for the mitral valve was 91% in P1, 96% in P2, and 97% in P3 (P1 vs. P2 $P = 0.008$; P1 vs. P3 $P = 0.006$; P2 vs. P3 $P = 0.195$). Early mitral valve replacement after initial repair were most often performed. MV-Replacement after MVR early: Mean interval 91 days; MV-Replacement after MVR late: Mean interval 927 days; MVR after MVR early: Mean interval 13.67 days; MVR after MVR late: Mean interval 317 days.

Survival rates (93% follow-up) were 66% (5-years), 55% (10 years), 44% (15 years) in P1, 63% (5 years), 50% (10 years) in P2, and 80% (5 years) in Period 3. General survival was 70% after 5 years, 56% after 10 years, and 45% after 15 years. 5-year survival was better in the recent

**Table 1. Demographic data.**

| | Period 1 | Period 2 | Period 3 | p-value total | p-value Period 1 vs. 2 | p-value Period 1 vs. 3 | p-value Period 2 vs. 3 |
|---|---|---|---|---|---|---|---|
| | 2004–2009 | 2010–2014 | 2015–2019 | | | | |
| **Age** | 64.26+/-11.27 | 66.46+/-12.1 | 65.42+/-11.47 | 0.046 | 0.04 | 0.323 | 0.323 |
| **Body Mass Index** | 26.46+/-4.19 | 26.8+/-4.65 | 27.11+/-4.66 | 0.582 | 1 | 1 | 1 |
| **EuroSCORE (ES)** | 6.52+/-2.91 | 7.92+/-3.74 | 6.71+/-3.32 | < 0.001 | < 0.001 | 0.857 | < 0.001 |
| **logistic ES** | 11.93+/-11.54 | 15.18+/-16.98 | 10.54+/-13.14 | < 0.001 | 0.263 | 0.216 | < 0.001 |
| **Hospitalization time** | 17.74+/-11.68 | 17.45+/-13.89 | 15.21+/-10.01 | 0.002 | 0.736 | 0.011 | 0.007 |
| **Male gender** | 58.21% [78] | 62.46% [183] | 63.55% [197] | 0.562 | 1 | 1 | 1 |
| **Primary MV-pathology** | 66% [92] | 55% [163] | 73% [230] | < 0.001 | 0.088 | 0.167 | < 0.001 |

**Table 2. Concomitant procedures.**

| | Period 1 | Period 2 | Period 3 | P Total | P Period 1 vs. 2 | P Period 1 vs. 3 | P Period 2 vs. 3 |
|---|---|---|---|---|---|---|---|
| | 2004–2009 | 2010–2014 | 2015–2019 | | | | |
| CABG | 34.33% [46] | 45.24% [133] | 27.42% [85] | < 0.001 | 0.088 | 0.176 | < 0.001 |
| Venous only CABG | 14.18% [19] | 25.51% [75] | 15.16% [47] | 0.001 | 0.025 | 0.903 | 0.007 |
| CABG plus IMA | 17.16% [23] | 17.69% [52] | 11.29% [35] | 0.063 | 1 | 0.251 | 0.102 |
| IMA only | 2.99% [4] | 2.04% [6] | 1.29% [4] | 0.399 | 1 | 1 | 1 |
| AV-Replacement | 14.93% [20] | 19.39% [57] | 23.23% [72] | 0.123 | 0.586 | 0.191 | 0.586 |
| AV-Repair | 0.75% [1] | 0.34% [1] | 0.97% [3] | 0.617 | 1 | 1 | 1 |
| MVR only | 47.76% [64] | 31.29% [92] | 39.68% [123] | 0.003 | 0.005 | 0.139 | 0.078 |
| MIS-MVR | 0% [0] | 3.74% [11] | 0.65% [2] | 0.004 | 0.105 | 0.873 | 0.058 |
| TVR/Replacement | 5.97% [8] | 11.9% [35] | 17.1% [53] | 0.005 | 0.171 | 0.009 | 0.171 |
| PFO/ASD | 2.24% [3] | 1.7% [5] | 1.94% [6] | 0.938 | 1 | 1 | 1 |
| VSD | 0% [0] | 1.02% [3] | 1.29% [4] | 0.54 | 1 | 1 | 1 |
| Aorta | 1.49% [2] | 2.72% [8] | 3.87% [12] | 0.43 | 1 | 0.922 | 1 |
| Electro | 0.75% [1] | 1.02% [3] | 1.61% [5] | 0.748 | 1 | 1 | 1 |
| MAZE/LAA | 3.73% [5] | 13.61% [40] | 10.97% [34] | 0.009 | 0.011 | 0.044 | 0.388 |
| MAZE only | 18.66% [25] | 14.97% [44] | 10.32% [32] | 0.046 | 0.412 | 0.072 | 0.22 |
| LAA only | 0% [0] | 0.68% [2] | 2.58% [8] | 0.06 | 0.47 | 0.393 | 0.393 |
| Re-OP | 5.22% [7] | 4.08% [12] | 2.58% [8] | 0.349 | 0.844 | 0.777 | 0.844 |
| IABP | 5.22% [7] | 10.54% [31] | 2.58% [8] | < 0.001 | 0.214 | 0.259 | < 0.001 |
| Carotis TEA | 0% [0] | 0.68% [2] | 0.32% [1] | 0.789 | 1 | 1 | 1 |
| LV-Aneurysm | 0% [0] | 0.34% [1] | 0% [0] | 0.58 | 1 | | 1 |

CABG = Coronary Artery Bypass Graft; IMA = Internal mammary artery; AV-Replacement = Aortic Valve Replacement, AV-Repair = Aortic Valve Repair; MVR = Mitral Valve Repair; MIS-MVR = Minimal-invasive MVR; TVR/Repair = Tricuspid Valve Replacement/repair; PFO/ASD = Persisting Foramen Ovale/Atrial Septal Defect; VSD = Ventricle Septal Defect; Aorta = Ascending Aorta wrapping or Replacement; Electro = Electrophysiological Implant; MAZE/LAA = Maze-Procedure (Atrial ablation) / Left Atrial Appendage; MAZE = Maze-Procedure (Atrial ablation); Re-OP = Redo-Operation; IABP = Intra aortic balloon pump; Carotis TEA = A. carotis interna thrombendarteriectomy; LV-Aneurysma = Left ventricular aneurysm

period than in both previous periods. The differences between Period 1 and 2, Period 1 and 3, and Period 2 and 3 were significant (p<0.001).

Odds ratios for reduced long-term survival were concomitant venous bypass surgery (5-years 1.871, p = 0.004; 10 years 2,701, p = 0.026); concomitant tricuspid valve surgery (10-years 4.699, p = 0.04). Survival was positively influenced by isolated MVR (5-years 0.375, p<0.001; 10-years 0.246, p = 0.001), concomitant isolated arterial bypass (IMA) (10-years 0.153, p0.051); posterior leaflet procedures (5 years 0.568, p = 0.005; 10 years 0.178, p<0.001), and use of artificial chordae (5-years 0.235, p<0.001) (Table 5, Fig 2).

## Discussion

Ever since Carpentier has propagated the mandatory use of an annuloplasty-ring in order to stabilize the reconstruction result it became the procedure the overwhelming majority of all surgeons adhered to owing to the clear and steadily growing body of evidence regarding the benefit of the ring [1–3]. Indeed, our study mirrored this indicating that ring implantation remained mandatory throughout the entire observation period. However, preference towards certain rings changed alongside improved designs. Those design changes comprised on one hand the very shape of the ring propagating the "saddle-shape" as being a design improvement with positive implications on hemodynamic performance but on the other hand on design

**Table 3. Types of additional repair maneuvers.**

|  | Period 1 | Period 2 | Period 3 | P Total | P Period 1 vs. 2 | P Period 1 vs. 3 | P Period 2 vs. 3 |
|---|---|---|---|---|---|---|---|
|  | 2004–2009 | 2010–2014 | 2015–2019 |  |  |  |  |
| AML | 11.94% [16] | 4.08% [12] | 5.16% [16] | 0.005 | 0.014 | 0.039 | 0.662 |
| PML | 54.48% [73] | 23.13% [68] | 21.29% [66] | < 0.001 | < 0.001 | < 0.001 | 0.656 |
| Plication | 16.42% [22] | 13.61% [40] | 20% [62] | 0,109 | 0,903 | 0,903 | 0.14 |
| Resection | 45.52% [61] | 15.31% [45] | 6.13% [19] | < 0.001 | < 0.001 | < 0.001 | < 0.001 |
| Artificial Chordae | 2.99% [4] | 22.45% [66] | 31.61% [98] | < 0.001 | < 0.001 | < 0.001 | 0.015 |
| Other Chordae Measures | 8.21% [11] | 1.36% [4] | 0% [0] | < 0.001 | 0,002 | < 0.001 | 0.119 |
| Wooler Plasty | 20.15% [27] | 2.04% [6] | 0.97% [3] | < 0.001 | < 0.001 | < 0.001 | 0.452 |
| Patch | 2.99% [4] | 1.02% [3] | 0.97% [3] | 0.207 | 0.749 | 0.749 | 1 |
| Ring-Refixation | 0% [0] | 0.68% [2] | 0.32% [1] | 0.789 | 1 | 1 | 1 |

AML = all measures on the anterior mitral leaflet; PML = all measures on the posterior mitral leaflet; Plication = Plication or inverse plication without resection; Resection: triangular or quadrangular resection on the leaflets. Artificial Chordae = Implantation of artificial chordae with stretched polytetrafluorethylen (Goretex™) from the papillary muscles to the AML and/or PML; Other Chordae Measures = other measures on the chordae or papillary muscles, i.e. repositioning, shortening, or resection. Wooler = Teflon-pledget-supported U-Suture on the anterior or posterior commissure; Patch = Implantation of a pericardium patch into the leaflet; Ring-Refixation = Refixationen of a partially detached annuloplasty-ring

changes intended to facilitate implantation itself [7–9]. These particular changes comprised wider rings with multi-layer fabric to allow for better stitching and more secure anchoring. Stiff or moderately flexible rings appeared to stabilize the annulus significantly better than highly flexible ones [10, 11] so that the use of the latter did not play a role in this study. Procedural characteristics for the valvular apparatus changed markedly from mainly classical

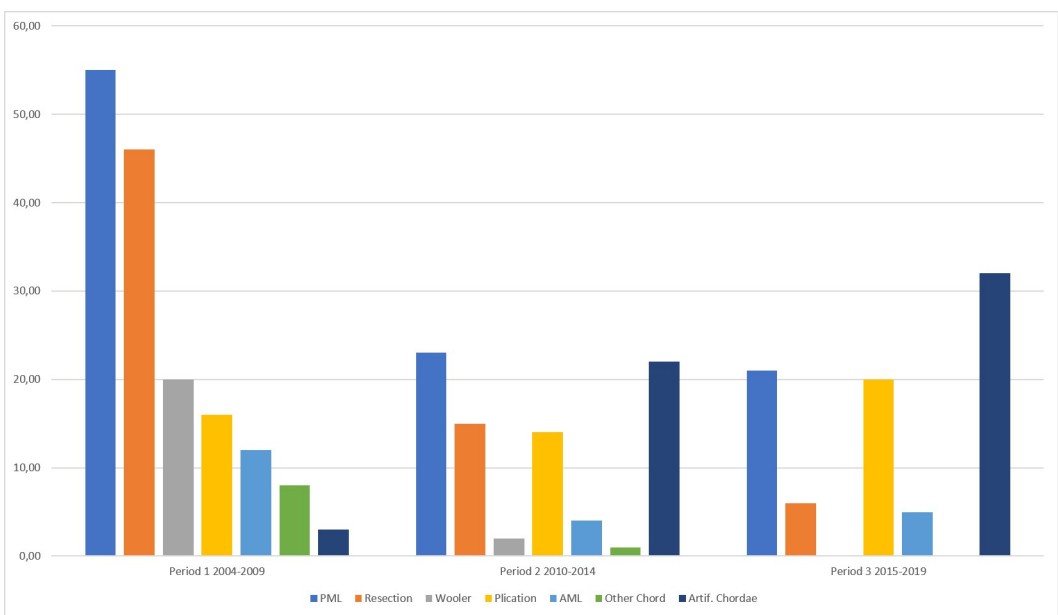

**Fig 1. Shift of additional reconstruction measures.** PML = Corrective measure on the posterior mitral leaflet; Resection = All types of resection measure on either leaflet; Wooler = Commissural reduction plasty according to Wooler; Plication = Plication on the leaflets without resection; AML = Corrective measure on the anterior mitral leaflet; Other Chord = Repositioning, shortening of the native chords, shortening of the origin of the chord by means of splitting or trimming or repositioning of the papillary muscle; Artif. Chordae = Chordae replacement with artificial chords.

**Table 4. Procedure-dependent early mortality.**

| | Early mortality respective treatment | Early mortality control | Odds ratio [mortality] | P |
|---|---|---|---|---|
| **Artificial Chordae** | 4.17% [7/168] | 12.98% [74/570] | 0.292 | < 0.001 |
| **Other Chordae Measures** | 0% [0/15] | 11.2% [81/723] | 0 | 0.393 |
| **CABG** | 17.8% [47/264] | 7.17% [34/474] | 2.799 | < 0.001 |
| **Venous CABG only** | 20.57% [29/141] | 8.71% [52/597] | 2.709 | < 0.001 |
| **CABG with IMA** | 15.45% [17/110] | 10.19% [64/628] | 1.61 | 0.135 |
| **IMA only** | 7.14% [1/14] | 11.05% [80/724] | 0.62 | 1 |
| **AV-Replacement** | 13.42% [20/149] | 10.36% [61/589] | 1.341 | 0.304 |
| **AV-Repair** | 20% [1/5] | 10.91% [80/733] | 2.038 | 0.442 |
| **MV-Replacement** | 11.11% [1/9] | 10.97% [80/729] | 1.014 | 1 |
| **MVR only** | 3.23% [9/279] | 15.69% [72/459] | 0.179 | < 0.001 |
| **MIS-MVR** | 0% [0/13] | 11.17% [81/725] | 0 | 0,38 |
| **TVR/Replacement** | 16.67% [16/96] | 10.12% [65/642] | 1.774 | 0.078 |
| **PFO/ASD** | 0% [0/14] | 11.19% [81/724] | 0 | 0.385 |
| **VSD** | 0% [0/7] | 11.08% [81/731] | 0 | 1 |
| **Aorta** | 9.09% [2/22] | 11.03% [79/716] | 0.807 | 1 |
| **Electro** | 55.56% [5/9] | 10.43% [76/729] | 10.671 | 0.001 |
| **MAZE/LAA** | 5.06% [4/79] | 11.68% [77/659] | 0.403 | 0.086 |
| **LAA only** | 0% [0/10] | 11.13% [81/728] | 0 | 0.612 |

CABG = Coronary Bypass; IMA = Internal mammary artery; AV-Replacement = Aortic Valve Replacement, AV-Repair = Aortic Valve Repair; MVR = Mitral Valve Repair; MIS-MVR = Minimal-invasive MVR; TVR/Replacement = Tricuspid valve repair/replacement; PFO/ASD = Persisting Foramen Ovale/Atrial Septal Defect; VSD = Ventricle Septal Defect; Aorta = Aortic wrapping or replacement; Electro = Electrophysiological implant; MAZE/LAA = Maze-Procedure (Atrial ablation) / Left Atrial Appendage

resection maneuvers to predominant use of artificial chordae [12–18]. In the early years the classical recommendations such as reshaping of the posterior leaflet by resection or plication maneuvers as well as reshaping of the annulus by appropriate rings were mostly followed but measures such as Wooler plasty, Paneth plasty, or Alfieri Stitch were still performed [19, 20]. However, they did not exhibit such robust evidence in the literature regarding a beneficial effect on long-term patency as posterior leaflet correction and/or artificial chordae. We did not perform the Paneth plasty, which are meant to reduce the diameter of the anulus but favored instead the annuloplasty-ring as an implant from the early days on. Alfieri-Stich was neither performed because of the, in our eyes, "non-anatomical" approach. The patients who had received a Wooler plasty, however, fared quite well and the presence of a Wooler plasty positively influenced long-term survival even after 10 years. This, however, was quite probably not due to the Wooler plasty itself but owing to the mandatory annuloplasty ring, these patients also received.

It cannot be ruled out, however, that intraoperative quality control also played a role regarding long-term stability. Transesophageal echocardiography (TEE) was not yet mandatory in period 1. Surgeons mainly relied on intraoperative filling of the left ventricle with saline to estimate the success of the reconstruction and no intraoperative quality control was thus performed while the heart was beating. Direct correction in case of an insufficient result was thus not an option [21–23]. The widespread introduction of artificial chordae was a relevant evolutionary step that our godfather Alain Carpentier had not foreseen. Perhaps this is because the most suitable material for these chordae, expanded polytetrafluoroethylene, Goretex ™, was not yet available in the 1970s. Artificial chordae were soon adopted by the majority of surgeons

**Table 5. Odds ratios of 5- and 10-year survival.**

| | 5-Year Survival | | 10-Year Survival | |
|---|---|---|---|---|
| | OR | P | OR | p |
| **Gender** | 0.86 [0.597–1.24] | 0.475 | 0.839 [0.49–1.435] | 0.612 |
| **Anterior Mitral leaflet** | 0.773 [0.352–1.699] | 0.654 | 0.286 [0.114–0.717] | 0.011 |
| **Posterior Mitral leaflet** | 0.355 [0.229–0.55] | 0 | 0.137 [0.077–0.243] | 0 |
| **Plication** | 0.46 [0.267–0.792] | 0.006 | 0.419 [0.213–0.826] | 0.018 |
| **Resection** | 0.461 [0.278–0.764] | 0.003 | 0.151 [0.083–0.277] | 0 |
| **Patch** | 0.492 [0.098–2.46] | 0.485 | 0.31 [0.043–2.243] | 0.245 |
| **Ring-Refixation** | 0.296 [0.014–6.193] | 0.518 | 1.602 [0.076–33.749] | 1 |
| **Wooler Plasty** | 0.612 [0.262–1.426] | 0.344 | 0.213 [0.094–0.479] | 0 |
| **Artificial Chordae** | 0.342 [0.199–0.589] | 0 | 1.725 [0.573–5.198] | 0.465 |
| **Other Chordae Measures** | 0.892 [0.211–3.777] | 1 | 0.307 [0.061–1.557] | 0.152 |
| **CABG** | 1.912 [1.325–2.758] | 0.001 | 2.17 [1.239–3.801] | 0.009 |
| **Venous CABG only** | 2.292 [1.481–3.549] | 0 | 2.358 [1.14–4.875] | 0.027 |
| **CABG plus IMA** | 1.253 [0.782–2.006] | 0.414 | 1.815 [0.843–3.909] | 0.171 |
| **IMA only** | 0.366 [0.077–1.743] | 0.329 | 0.204 [0.033–1.245] | 0.092 |
| **AV-Replacement** | 2.084 [1.326–3.274] | 0.002 | 1.76 [0.867–3.573] | 0.157 |
| **AV-Repair** | 1.495 [0.209–10.701] | 1 | 2.253 [0.115–44.117] | 1 |
| **MV-Replacement** | 0.741 [0.183–2.998] | 0.746 | 0.24 [0.063–0.918] | 0.039 |
| **MVR** | 0.663 [0.229–1.921] | 0.627 | 0.307 [0.039–2.438] | 0.47 |
| **MVR only** | 0.242 [0.156–0.375] | 0 | 0.202 [0.115–0.355] | 0 |
| **MIS-MVR** | 0.069 [0.004–1.176] | 0.007 | 0.313 [0.019–5.072] | 0.423 |
| **TVR/Replacement** | 2.363 [1.341–4.161] | 0.004 | 4.023 [1.198–13.507] | 0.014 |
| **PFO/ASD** | 0.892 [0.211–3.777] | 1 | 0.468 [0.077–2.853] | 0.597 |
| **VSD** | 0.744 [0.067–8.258] | 1 | 0.957 [0.039–23.748] | 1 |
| **Aorta** | 0.532 [0.167–1.696] | 0.423 | 1.921 [0.228–16.22] | 1 |
| **Electro** | 19.951 [1.118–356.183] | 0.004 | 4.239 [0.236–76.139] | 0.342 |
| **MAZE/LAA** | 0.743 [0.408–1.353] | 0.409 | 2.212 [0.638–7.665] | 0.309 |
| **LAA only** | 0.245 [0.029–2.047] | 0.251 | 0.957 [0.039–23.748] | 1 |

CABG = Coronary Bypass; IMA = Internal mammary artery; AV-Replacement = Aortic Valve Replacement, AV-Repair = Aortic Valve Repair; MVR = Mitral Valve Repair; MIS-MVR = Minimal-invasive MVR; TVR/Replacement = Tricuspid valve repair/replacement; PFO/ASD = Persisting Foramen Ovale/Atrial Septal Defect; VSD = Ventricle Septal Defect; Aorta = Aortic wrapping or replacement; Electro = Electrophysiological implant; MAZE/LAA = Maze-Procedure (Atrial ablation) / Left Atrial Appendage

and gained worldwide acceptance. Indeed, that was apparent also in our cohorts over the years. Although early and intermediate results up to five years were quite favorable for valves repaired with artificial chordae this benefit disappeared in the long-term results (10-year survival). It must be noted, however, that artificial chordae came into wider use but in the second and third period so that long-term results on a larger scale are not yet available.

We noticed a steep increase in repair from period 1 to period 2 possibly owing to the experience gained with repair strategies in the early years in our institution. From period 2 to period 3 a slight decrease of the proportion of repair maneuvers was noted declining from 48% to 41%. This shift was not due to a more restrictive policy regarding repair surgery but was instead non-intentional and can be explained by a higher morbidity of the patients resulting in a higher proportion of patients with complex redo-surgery or endocarditis.

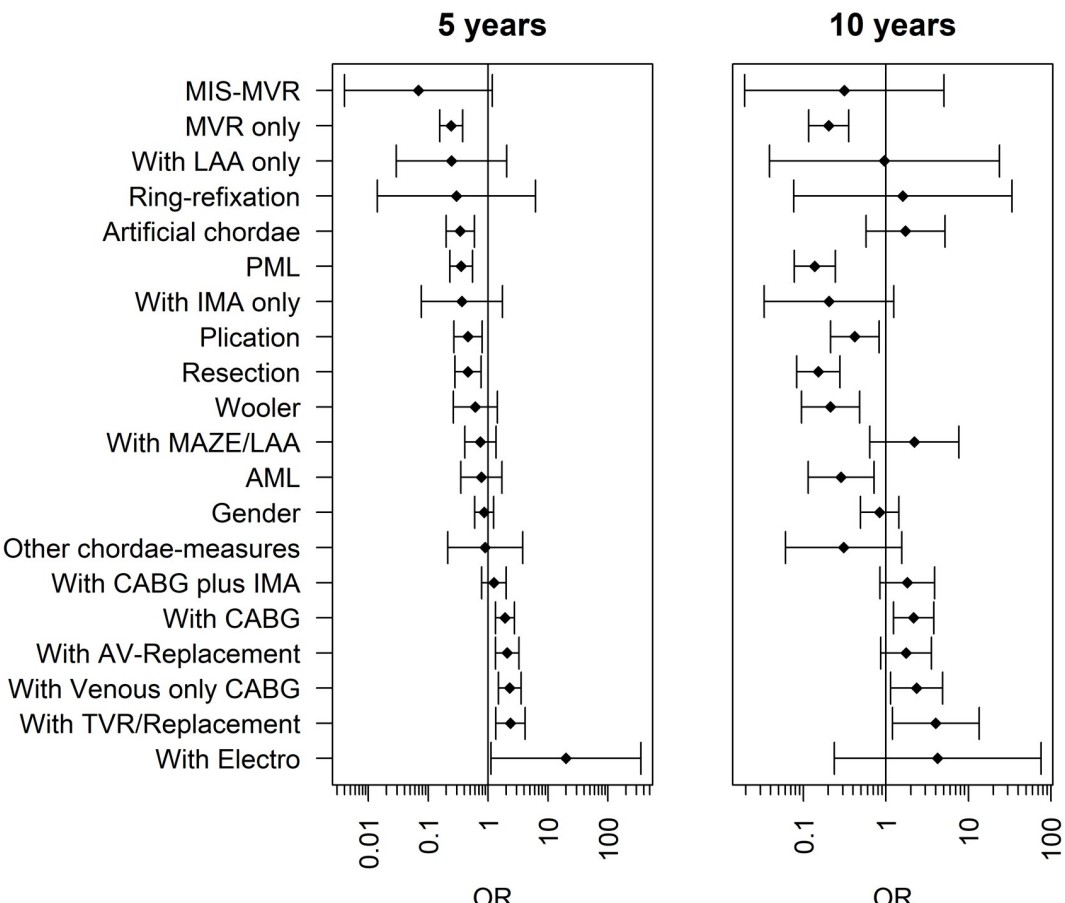

**Fig 2. Odds ratios for long-term survival (5- and 10 years).** MIC = Minimal invasive; MVR = Mitral valve repair; LAA = Left atrial appendage closure; PML = posterior mitral leaflet; IMA = Internal mammary artery; MAZE = Atrial ablation therapy; AML = Anterior mitral leaflet; CABG = Coronary artery bypass graft; AV-Replacement = Aortic valve replacement; TVR/Replacement = Tricuspid valve repair or replacement; Electro = Electrophysiological device.

The increase in patients with primary mitral regurgitation from period 1 to period 3 cannot be readily explained. It is conceivable that, because of demographic change, patients with restrictive mitral valve pathology presented with increasing multimorbidity, so that the indication for repair in patients with secondary mitral valve pathology became more defensive.

Only 13 minimally invasive mitral valve repair procedures were carried out mainly in period 2. The true minimally invasive approach was indeed not favored in our clinic until very recently because of lack of center-wide, multi-surgeon expertise.

Leaflet repair strategies especially concerning the posterior leaflet remained a frequent procedure demonstrating excellent results even in the long-term. Why is it so? The posterior leaflet has a much less complex movement and serves mainly as an abutment for the very mobile anterior leaflet. Thus, a stable coaptation for the anterior leaflet can be rather easily achieved with appropriate measures on the posterior leaflet such as quadrangular resection, plication or cleft-closure accompanied by remodeling the often v-shaped distorted posterior anulus back to its distinct anteriorly flattened circle, the "smiley" mouth. In contrast, any reshaping measure of the highly mobile anterior leaflet is much more prone to frustrating results (Fig 3). While co-morbidities negatively influenced survival early as well as late [24, 25], it was interesting to see that patients with concomitant venous only coronary bypass surgery fared worse than the others (Fig 3).

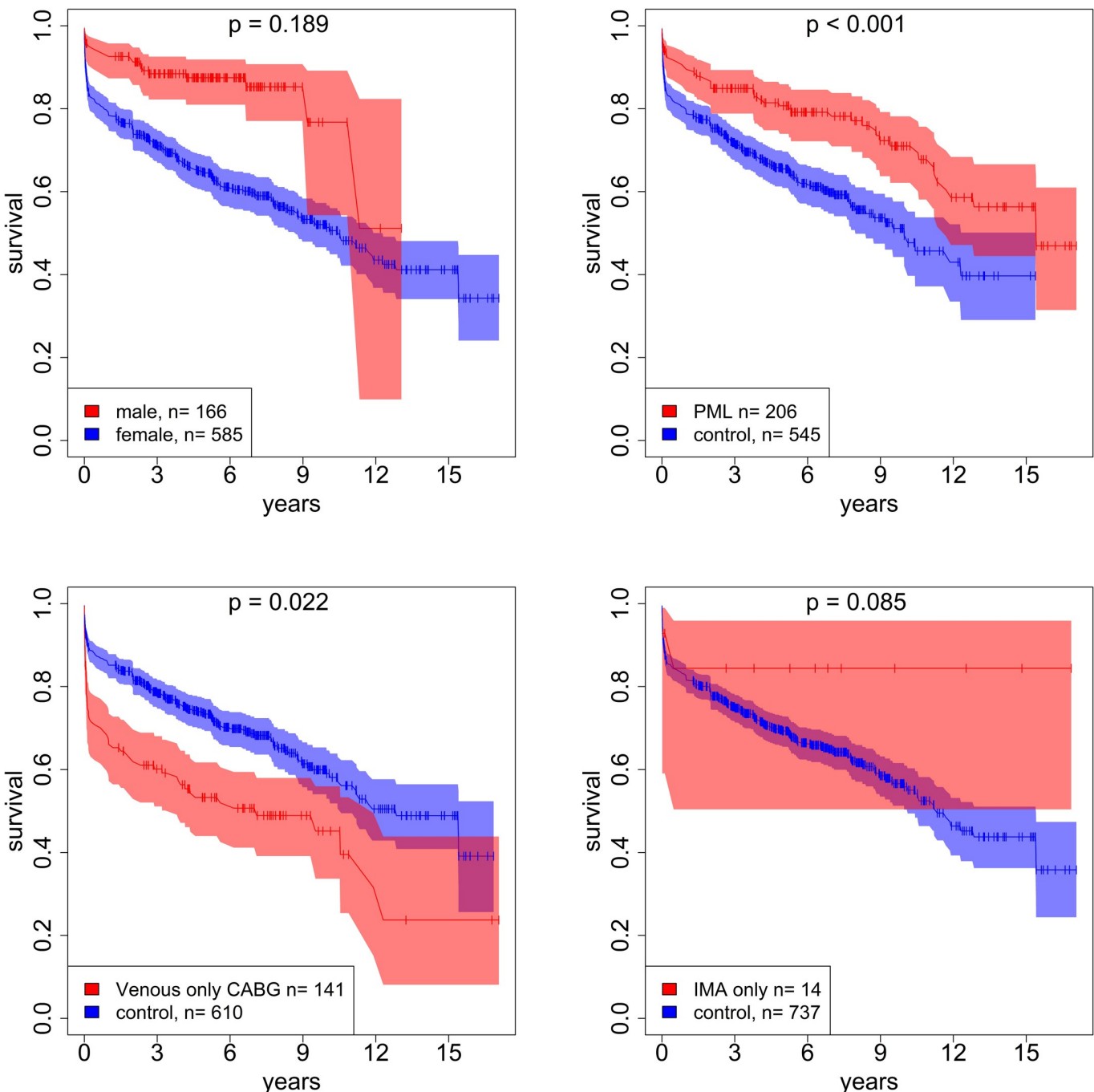

**Fig 3. Procedure-related survival.** MVR = Mitral valve repair; Resection = All types of resection measure on either leaflet; CABG = Coronary artery Bypass Graft; IMA = Internal mammary artery.

This can, however, be explained. These patients were too sick to be deemed fit for the use of a mammary artery in the eyes of the surgeon carrying out the procedure. Such a surgical decision was made upon the general clinical appraisal of the patient and not concrete facts such as the EuroSCORE. Thus, the underlying reasons for such a decision remained somehow elusive. This finding gains additional validation when looking at the fate of the patients who indeed

had received LIMA only bypass surgery. They fared significantly better than the control. These patients only had a 1-vessel disease and received an artery for prognostic reasons, i.e. profiting from it because of a better life expectancy in the eyes of the surgeon (Fig 3). At the other end of the spectrum were those patients receiving aortic valve replacement, concomitant tricuspid valve repair or even replacement or those requiring electrophysiological implants. These patients fared worse early as well as late indicating the higher morbidity at the time of surgery.

In general, early mortality risks were typically dependent on the patient's individual situation. Procedural success and stability increased over the three periods and long-term results were very acceptable. The patients of period 3 demonstrated a superior 5-year survival than that of the two previous groups. Whether this is due to a more precise indication and/or better perioperative management in this latter period is speculative. It remains to be seen whether this trend will continue significantly after 10 or 15 years, respectively (Fig 4).

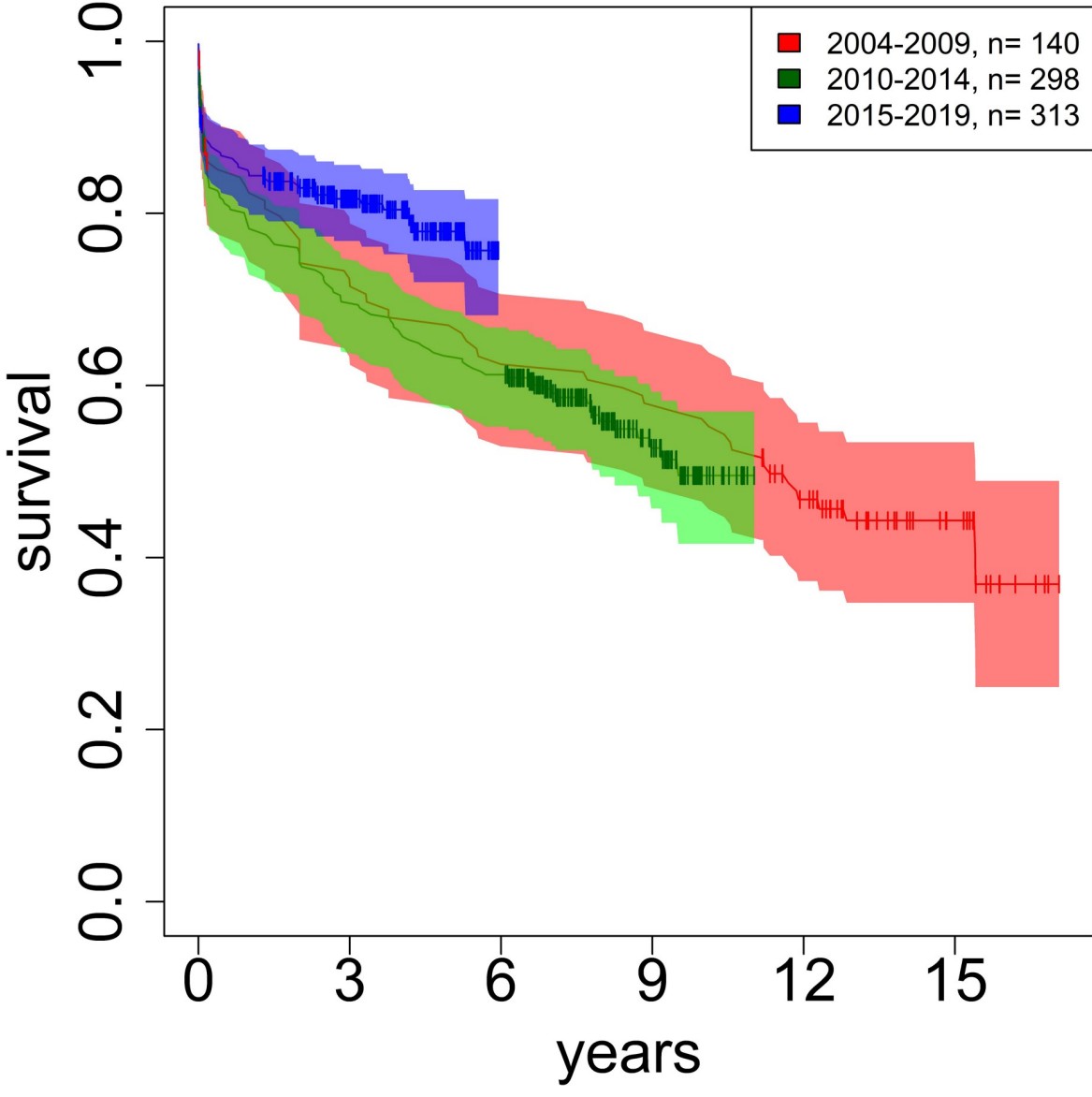

**Fig 4. Cohort-Survival.**

## Limitations

This study is of retrospective nature and thus naturally subject to the typical limitations. Mitral valve pathologies could only be roughly obtained. In almost all cases, however, we were able to differentiate between primary or secondary nature from the still available documents (surgery, echocardiography). The procedures were carried out upon the discretion of the respective surgeon and not prospectively assigned in a randomized fashion. Long-term follow-up was not complete. We tried best to reach all patients but particularly in the rural regions the whereabouts of some patients remained elusive despite all efforts. We used many different rings in the last 15 years. However, not all of the available rings were used at respective times. Thus, the selection is incomplete and is even somehow biased as the clinics policy always was and is to reduce the number of implants to a reasonable minimum for budgetary purposes.

## Conclusion

Mitral valve repair underwent a qualitative evolution in the last 15 years. It became clear that the use of a ring as well as artificial chordae and appropriate reshaping of the leaflets, particularly the posterior one, were keys to success. Thus, proving that the combination of the best well established old-fashioned techniques such as leaflet resection measures and mandatory use of rings as well as the most useful modern approaches such as artificial chordae and modern saddle-shaped ring designs is in the best interest of the patient. Co-morbidities and thus, the necessity to perform concomitant procedures in more than half of the entire cohort markedly influenced survival. However, almost half of the patients are alive after 15 years. Mitral valve repair is one of the few remaining primary cardiosurgical procedures and it will remain in our hands as long as we are able to provide a reliable, stable and long-lasting anatomical repair surpassing the results of any interventional strategy [26–30]. Luckily, the complexity of the mitral valves plays into our cards in this regard. Thus, mitral valve repair is still and for the years to come one of the most valuable cardiosurgical procedures.

## Acknowledgments

The authors thank Sofia Chopsonidou for helping with data acquisition.

## Author Contributions

**Conceptualization:** Johannes M. Albes.

**Data curation:** Martin Hartrumpf.

**Formal analysis:** Filip Schröter.

**Investigation:** Farnoosh Motazedian.

**Methodology:** Johannes M. Albes.

**Resources:** Johannes M. Albes.

**Software:** Martin Hartrumpf.

**Supervision:** Roya Ostovar, Johannes M. Albes.

**Validation:** Roya Ostovar, Martin Hartrumpf.

**Visualization:** Filip Schröter.

**Writing – original draft:** Farnoosh Motazedian.

**Writing – review & editing:** Johannes M. Albes.

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
