## [Decision Letter · Decision Letter 0]

18 Feb 2022

PONE-D-21-35308Everyday mitral valve reconstruction. What has changed over the last 15 years?PLOS ONE

Dear Dr. Albes

Thank you for submitting your manuscript to PLOS ONE. After careful consideration, we feel that it has merit but does not fully meet PLOS ONE’s publication criteria as it currently stands. Therefore, we invite you to submit a revised version of the manuscript that addresses the points raised during the review process.

We look forward to receiving your revised manuscript.

Kind regards,

Alessandro Leone, MD

Academic Editor

PLOS ONE

Journal Requirements:

3. Please provide additional details regarding participant consent. In the ethics statement in the Methods and online submission information, please ensure that you have specified what type you obtained (for instance, written or verbal, and if verbal, how it was documented and witnessed). If your study included minors, state whether you obtained consent from parents or guardians. If the need for consent was waived by the ethics committee, please include this information

4. We note that you have stated that you will provide repository information for your data at acceptance. Should your manuscript be accepted for publication, we will hold it until you provide the relevant accession numbers or DOIs necessary to access your data. If you wish to make changes to your Data Availability statement, please describe these changes in your cover letter and we will update your Data Availability statement to reflect the information you provide

Please review your reference list to ensure that it is complete and correct. If you have cited papers that have been retracted, please include the rationale for doing so in the manuscript text, or remove these references and replace them with relevant current references. Any changes to the reference list should be mentioned in the rebuttal letter that accompanies your revised manuscript. If you need to cite a retracted article, indicate the article’s retracted status in the References list and also include a citation and full reference for the retraction notice

Reviewers' comments:

Reviewer's Responses to Questions

**Comments to the Author**

1. Is the manuscript technically sound, and do the data support the conclusions?

Reviewer #1: Yes

Reviewer #2: Yes

2. Has the statistical analysis been performed appropriately and rigorously? 

Reviewer #1: Yes

Reviewer #2: Yes

3. Have the authors made all data underlying the findings in their manuscript fully available?

Reviewer #1: Yes

Reviewer #2: Yes

4. Is the manuscript presented in an intelligible fashion and written in standard English?

Reviewer #1: Yes

Reviewer #2: Yes

5. Review Comments to the Author

Reviewer #1: Dear the authors of the manuscript entitled "Everyday mitral valve reconstruction. What has changed over the last 15 years?"

Thank you for writing this retrospective study which evaluates the change in the paradigm of mitral valve reconstruction over a three periods of time

I valued the way you presented this study and the results and discussion parts

I think this study provided outcomes after mitral valve reconstruction and also provided the changes in the method of surgical techniques utilized

One major issue which I believe is adding the mitral valve pathologies in these patients, which I think is very related to the outcomes studied

Otherwise I have no concerns

Thank you

Reviewer #2: This paper describes the evolution of mitral valve repair over many year. The evolution peroid was divided into three phases.

I have the following notice:

1. In the results in line 100-102 the was a decrease in MVR in P3. Can you explain it please?

2. Only 13 minimally invasive MVR?

3. Some of the abbreviation are known German abbreviation used with English meaning like MIC (minimal invasive Chirurgie) for minimally invasive. Could you please correct this?

4. From line 200 to 203: you wrote. (These particular changes 201 comprised wider rings with multi-layer fabric to allow for better stitching and more secure 202 anchoring. Stiff or moderately flexible rings appeared to stabilize the annulus significantly 203 better than highly flexible ones so that the use of the latter did not play a role in this study). Can you please give an evidence or a reference to this sentence.!

In general, the paper proves the thought of cardiac surgeons the use of artificial chordae is dominating the field of mitral surgery. The only unexplainable finding is he very low number of minimally invasive procedure which could be a center decision to avoid such an approach but with still no explanation. This nice collection needs some improvement before publication beside a language editing.

Best regards

6. PLOS authors have the option to publish the peer review history of their article (what does this mean?). If published, this will include your full peer review and any attached files.

Reviewer #1: **Yes: **Salah Eldien Altarabsheh

Reviewer #2: No

---

## [Author Response · Author response to Decision Letter 0]

15 Mar 2022

Dear Reviewers,

Thank you for your thoughtful comments and suggestions, which we would like to respond to as follows:

Reviewer #1: 

Mitral valve pathologies are indeed important. I am sure you know from your own experience that it is somewhat problematic to determine the exact underlying pathology in a retrospective study with an almost historical cohort going back to 2004, with few meaningful echocardiographic findings properly documented or with not much in the surgeon's protocol or in the letter from the referring hospital. Thus, pathologies were not the primary focus of interest in this retrospective study. However, on your advice, we reviewed all the data still available and obtained as much information as possible. In the end, we were able to distinguish quite well between primary and secondary nature of the origin of mitral valve regurgitation by considering the morphology as primary origin if the valve leaflets and other aspects directly affected the valve apparatus or as secondary origin if the valves were tethered and/or the annulus was asymmetrically distorted. In the remaining cases, whose pathological origin was still unclear, we assumed that it was a secondary condition if bypass surgery was performed at the same time. Indeed, our results show differences between patients with measures on the valves themselves (resection, reconstruction, repair) and those with concomitant bypass surgery. In this way, we were able to investigate, at least indirectly, the importance of the type of mitral valve regurgitation (primary or secondary origin) on outcome. We included mitral valve pathologies in Table 1 and referred to them in the Patients and Methods section (lines 72-5) and in the Results (line 107), Discussion (245-9), and Limitations (lines 291-3). We also cited the 2021 ESC guidelines, which contain the current mitral valve pathology systematics [5].

Reviewer #2: 

1. The slight decrease between period 2 and 3 was indeed apparent. After noticing a steep increase in repair from period 1 to period 2 possibly owing to the experience gained with repair strategies in the early years in our institution a slight decrease of the proportion of repair maneuvers was noted declining from 48% to 41%. This shift was not due to a more restrictive policy regarding repair surgery but was instead non-intentional and can be explained by a higher morbidity of the patients resulting in a higher proportion of patients with complex redo-surgery or endocarditis. We explained the slight decrease of the proportion of mitral valve repair and added a respective statement in the Discussion (lines 239-44).

2. Only 13 minimally invasive mitral valve repairs were performed mainly in period 2. The true minimally invasive approach was not favored in our hospital until recently because of the lack of center-wide expertise performed by multiple surgeons. So, you are correct. This was a center decision at the time, which has now been changed after several experienced surgeons had been trained in the meantime with minimally invasive mitral valve repair with left mini-thoracotomy at other institutions with high experience and volume. We added a respective statement in the discussion (lines 250-2) 

3. We have erased the term MIC in tables, legends, and figure 2 and replaced it by MIS-MVR. 

4. We added respective literature evidence regarding our statement: These particular changes comprised wider rings with multi-layer fabric to allow for better stitching and more secure anchoring. Stiff or moderately flexible rings appeared to stabilize the annulus significantly better than highly flexible ones [10,11].

Thank you again for your valuable comments.

Johannes M. Albes, MD, PhD, MBA

---

## [Decision Letter · Decision Letter 1]

24 May 2022

Everyday mitral valve reconstruction. What has changed over the last 15 years?

PONE-D-21-35308R1

Dear Dr.Johannes Prof. Dr. Albes,

We’re pleased to inform you that your manuscript has been judged scientifically suitable for publication and will be formally accepted for publication once it meets all outstanding technical requirements.

Kind regards,

Chengming Fan, MD, PhD

Academic Editor

PLOS ONE

Additional Editor Comments (optional):

Reviewers' comments:

Reviewer's Responses to Questions

**Comments to the Author**

1. If the authors have adequately addressed your comments raised in a previous round of review and you feel that this manuscript is now acceptable for publication, you may indicate that here to bypass the “Comments to the Author” section, enter your conflict of interest statement in the “Confidential to Editor” section, and submit your "Accept" recommendation.

Reviewer #1: All comments have been addressed

Reviewer #2: All comments have been addressed

2. Is the manuscript technically sound, and do the data support the conclusions?

Reviewer #1: Yes

Reviewer #2: Yes

3. Has the statistical analysis been performed appropriately and rigorously? 

Reviewer #1: Yes

Reviewer #2: Yes

4. Have the authors made all data underlying the findings in their manuscript fully available?

Reviewer #1: Yes

Reviewer #2: Yes

5. Is the manuscript presented in an intelligible fashion and written in standard English?

Reviewer #1: Yes

Reviewer #2: Yes

6. Review Comments to the Author

Reviewer #1: Dear the authors

I want to thank you for taking into consideration for my as well as the reviewer's comments which are all meant to improve the quality of the manuscript

I have no concerns about the manuscript in its current status

Reviewer #2: no comment more needed to be added. the authors has adressed all our concerns in the revised version of the paper

7. PLOS authors have the option to publish the peer review history of their article (what does this mean?). If published, this will include your full peer review and any attached files.

Reviewer #1: **Yes: **Salah Eldien Altarabsheh

Reviewer #2: No

---

## [Editor Report · Acceptance letter]

20 Jun 2022

PONE-D-21-35308R1 

Every day mitral valve reconstruction: What has changed over the last 15 years? 

Dear Dr. Albes:

I'm pleased to inform you that your manuscript has been deemed suitable for publication in PLOS ONE. Congratulations! Your manuscript is now with our production department. 

Kind regards, 

on behalf of

Dr. Chengming Fan 

Academic Editor

PLOS ONE